# Reading Order Matters: Information Extraction from Visually-rich Documents by Token Path Prediction

**Chong Zhang**[1][†][*], **Ya Guo**[2][†], **Yi Tu**[2], **Huan Chen**[2], **Jinyang Tang**[2],
**Huijia Zhu**[2], **Qi Zhang**[1][‡], **Tao Gui**[3][‡]

[1]School of Computer Science, Fudan University, Shanghai, China
[2]Ant Info Security Lab, Ant Group Inc., Hangzhou, China
[3]Institute of Modern Languages and Linguistics, Fudan University, Shanghai, China
{chongzhang20, qz, tgui}@fudan.edu.cn
{guoya.gy,qianyi.ty,chenhuan.chen,jinyang.tjy,huijia.zhj}@antgroup.com

## Abstract

Recent advances in multimodal pre-trained models have significantly improved information extraction from visually-rich documents (VrDs), in which named entity recognition (NER) is treated as a sequence-labeling task of predicting the BIO entity tags for tokens, following the typical setting of NLP. However, BIO-tagging scheme relies on the correct order of model inputs, which is not guaranteed in real-world NER on scanned VrDs where text are recognized and arranged by OCR systems. Such reading order issue hinders the accurate marking of entities by BIO-tagging scheme, making it impossible for sequence-labeling methods to predict correct named entities. To address the reading order issue, we introduce Token Path Prediction (TPP), a simple prediction head to predict entity mentions as token sequences within documents. Alternative to token classification, TPP models the document layout as a complete directed graph of tokens, and predicts token paths within the graph as entities. For better evaluation of VrD-NER systems, we also propose two revised benchmark datasets of NER on scanned documents which can reflect real-world scenarios. Experiment results demonstrate the effectiveness of our method, and suggest its potential to be a universal solution to various information extraction tasks on documents.

## 1 Introduction

Visually-rich documents (VrDs), including forms, receipts and contracts, are essential tools for gathering, carrying, and displaying information in the digital era. The ability to understand and extract information from VrDs is critical for real-world applications. In particular, the recognition

(a) Entity annotation by BIO-tags. With disordered inputs, BIO-tags fail to assign a proper label for each word to mark entities clearly.

(b) Entity annotation as word sequences within the document. The entity annotations are not affected by model input order.

Figure 1: A document image with its layout and entity annotations. (a) Sequence-labeling methods are not suitable for VrD-NER as model inputs are disordered in real situation. (b) We address the reading order issue by treating VrD-NER as predicting word sequences within documents.

and comprehension of scanned VrDs are necessary in various scenarios, including legal, business, and financial fields (Stanisławek et al., 2021; Huang et al., 2019; Stray and Svetlichnaya, 2020). Recently, several transformer-based multimodal pre-trained models (Garncarek et al., 2020; Xu et al., 2021a; Li et al., 2021a; Hong et al., 2022; Huang et al., 2022; Tu et al., 2023) have been proposed. Known as document transformers, these models can encode text, layout, and image inputs into a unified feature representation, typically by marking each input text token with its xy-coordinate on the document layout using an additional 2D positional embedding. Thus, document transformers can adapt to a wide range of VrD tasks, such as Named

---

[*]This work was done when the author was an intern at Ant Group.
[†]Equal contribution.
[‡]Corresponding author.

Entity Recognition (NER) and Entity Linking (EL) (Jaume et al., 2019; Park et al., 2019).

However, in the practical application of information extraction (IE) from scanned VrDs, **the reading order issue** is known as a pervasive problem that lead to suboptimal performance of current methods. This problem is particularly typical in VrD-NER, a task that aims to identify word sequences in a document as entities of predefined semantic types, such as headers, names and addresses. Following the classic settings of NLP, current document transformers typically treat this task as a sequence-labeling problem, tagging each text token using the BIO-tagging scheme (Ramshaw and Marcus, 1999) and predicting the entity tag for tokens through a token classification head. These sequence-labeling-based methods assumes that each entity mention is a **continuous and front-to-back** word sequence within inputs, which is always valid in plain texts. However, for scanned VrDs in real-world, where text and layout annotations are recognized and arranged by OCR systems, typically in a top-to-down and left-to-right order, the assumption may fail and the reading order issue arises, rendering the incorrect order of model inputs for document transformers, and making these sequence-labeling methods inapplicable. For instance, as depicted in Figure 1, the document contains two entity mentions, namely *"NAME OF ACCOUNT"* and *"# OF STORES SUPPLIED"*. However, due to their layout positions, the OCR system would recognize the contents as three segments and arrange them in the following order:(1)*"# OF STORES"*; (2)*"NAME OF ACCOUNT"*; (3)*"SUPPLIED"*. Such disordered input leads to significant confusion in the BIO-tagging scheme, making the models unable to assign a proper label for each word to mark the entity mentions clearly. Unfortunately, the reading order issue is particularly severe in documents with complex layouts, such as tables, multi-column contents, and unaligned contents within the same row or column, which are quite common in scanned VrDs. Therefore, we believe that the sequence-labeling paradigm is not a practical approach to address NER on scanned VrDs in real-world scenarios.

To address the reading order issue, we introduce Token Path Prediction (TPP), a simple yet strong prediction head for VrD-IE tasks. TPP is compatible with commonly used document transformers,

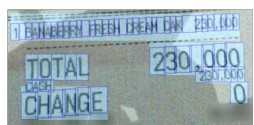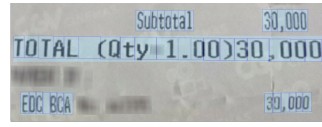

Figure 2: The reading order issue of scanned documents in real-world scenarios. Left: The entity *"TOTAL 230,000"* is disordered when reading from top to down. The entity *"CASH CHANGE"* lies in multiple rows and is separated to two segments. Right: The entity *"TOTAL 30,000"* is interrupted by *"(Qty 1.00)"* when reading from left to right. All these situations result in disordered model inputs and affect the performance of sequence-labeling based VrD-NER methods. More real-world examples of disordered layouts are displayed in Figure 7 in Appendix.

and can be adapted to various VrD-IE and VrD understanding (VrDU) tasks, such as NER, EL, and reading order prediction (ROP). Specifically, when using TPP for VrD-NER, we model the token inputs as a complete directed graph of tokens, and each entity as a token path, which is a group of directed edges within the graph. We adopt a grid label for each entity type to represent the token paths as $n * n$ binary values of whether two tokens are linked or not, where $n$ is the number of text tokens. Model learns to predict the grid labels by binary classification in training, and search for token paths from positively-predicted token pairs in inference. Overall, TPP provides a viable solution for VrD-NER by presenting a suitable label form, modeling VrD-NER as predicting token paths within a graph, and proposes a straightforward method for prediction. This method does not require any prior reading order and is therefore unaffected by the reading order issue. For adaptation to other VrD tasks, TPP is applied to VrD-ROP by predicting a global path of all tokens that denotes the predicted reading order. TPP is also capable of modeling the entity linking relations by marking linked token pairs within the grid label, making it suitable for direct adaptation to the VrD-EL task. Our work is related to discontinuous NER in NLP as the tasks share a similar form, yet current discontinuous NER methods cannot be directly applied to address the reading order issue since they also require a proper reading order of contents.

For better evaluation of our proposed method, we also propose two revised datasets for VrD-NER. In current benchmarks of NER on scanned VrDs, such as FUNSD (Jaume et al., 2019) and CORD

(Park et al., 2019), the reading order is manually-corrected, thus failing to reflect the reading order issue in real-world scenarios. To address this limitation, we reannotate the layouts and entity mentions of the above datasets to obtain two revised datasets, FUNSD-r and CORD-r, which accurately reflect the real-world situations and make it possible to evaluate the VrD-NER methods in disordered scenarios. We conduct extensive experiments by integrating the TPP head with different document transformer backbones and report quantitative results on multiple VrD tasks. For VrD-NER, experiments on the FUNSD-r and CORD-r datasets demonstrate the effectiveness of TPP, both as an independent VrD-NER model, and as a pre-processing mechanism to reorder inputs for sequence-labeling models. Also, TPP achieves SOTA performance on benchmarks for VrD-EL and VrD-ROP, highlighting its potential as a universal solution for information extraction tasks on VrDs. The main contribution of our work are listed as follows:

1. We identify that sequence-labeling-based VrD-NER methods is unsuitable for real-world scenarios due to the reading order issue, which is not adequately reflected by current benchmarks.

2. We introduce Token Path Prediction, a simple yet strong approach to address the reading order issue in information extraction on VrDs.

3. Our proposed method outperforms SOTA methods in various VrD tasks, including VrD-NER, VrD-EL, and VrD-ROP. We also propose two revised VrD-NER benchmarks reflecting real-world scenarios of NER on scanned VrDs.

## 2   Related Work

**Sequence-labeling based NER with Document Transformers** Recent advances of pre-trained techniques in NLP (Devlin et al., 2019; Zhang et al., 2019) and CV (Dosovitskiy et al., 2020; Li et al., 2022c) have inspired the design of pre-trained representation models in document AI, in which document transformers (Xu et al., 2020, 2021a; Huang et al., 2022; Li et al., 2021a,c; Hong et al., 2022; Wang et al., 2022; Tu et al., 2023) are proposed to act as a layout-aware representation model of document in various VrD tasks. By the term document transformers, we refer to transformer encoders that take vision (optional), text and layout input of a document as a token sequence, in which each text token is embedded

together with its layout. In sequence-labeling based NER with document transformers, BIO-tagging scheme assigns a BIO-tag for each text token to mark entities: *B-ENT/I-ENT* indicates the token to be the beginning/inside token of an entity with type *ENT*, and *O* indicates the word does not belong to any entity. In this case, the type and boundary of each entity is clearly marked in the tags. The BIO-tags are treated as the classification label of each token and is predicted by a token classification head of the document transformer. As illustrated in introduction, sequence-labeling methods are typical for NER in NLP and is adopted by current VrD-NER methods, but is not suitable for VrD-NER in real-world scenarios due to the reading order issue.

**Reading-order-aware Methods** Several studies have addressed the reading order issue on VrDs in two directions: (1) Task-specific models that directly predict the reading order, such as LayoutReader, which uses a sequence-to-sequence approach (Wang et al., 2021b). However, this method is limited to the task at hand and cannot be directly applied to VrD-IE tasks. (2) Pre-trained models that learn from supervised signals during pre-training to improve their awareness of reading order. For instance, ERNIE-Layout includes a pre-training objective of reading order prediction (Peng et al., 2022); XYLayoutLM enhances the generalization of reading order pre-training by generating various proper reading orders using an augmented XY Cut algorithm (Gu et al., 2022). However, these methods require massive amount of labeled data and computational resources during pre-training. Comparing with the above methods, our work is applicable to multiple VrD-IE tasks of VrDs, and can be integrated with various document transformers, without the need for additional supervised data or computational costs.

**Discontinuous NER** The form of discontinuous NER is similar to VrD-NER, as it involves identifying discontinuous token sequences from text as named entities. Current methods of discontinuous NER can be broadly categorized into four groups: (1) Sequence-labeling methods with refined BIO-tagging scheme (Tang et al., 2015; Dirkson et al., 2021), (2) 2D grid prediction methods (Wang et al., 2021a; Li et al., 2022b; Liu et al., 2022), (3) Sequence-to-sequence methods (Li et al., 2021b; He and Tang, 2022), and (4) Transition-based methods (Dai et al., 2020). These methods all rely

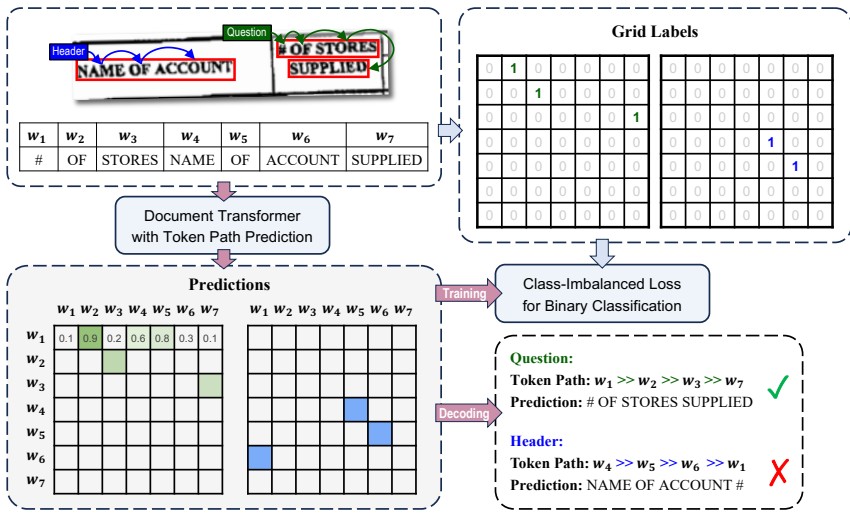

Figure 3: An overview of the training procedure of Token Path Prediction for VrD-NER. A TPP head for each entity type predicts whether an input tokens links to an another. The predict result is viewed as an adjacent matrix in which token paths are decoded as entity mentions. The overall model is optimized by the class-imbalance loss.

on a proper reading order to predict entities from front to back, thus cannot be directly applied to address the reading order issue in VrD-NER.

## 3 Methodology

### 3.1 VrD-NER

The definition of NER on visually-rich documents with layouts is formalized as follows. A visually-rich document with $N_\mathcal{D}$ words is represented as $\mathcal{D} = \{(w_i, \mathbf{b}_i)\}_{i=1,\dots,N_\mathcal{D}}$, where $w_i$ denotes the $i$-th word in document and $\mathbf{b}_i = (x_i^0, y_i^0, x_i^1, y_i^1)$ denotes the position of $w_i$ in the document layout. The coordinates $(x_i^0, y_i^0)$ and $(x_i^1, y_i^1)$ correspond to the bottom-left and top-right vertex of $w_i$'s bounding box, respectively. The objective of VrD-NER is to predict all the entity mentions $\{s_1, \dots, s_J\}$ within document $\mathcal{D}$, given the pre-defined entity types $\mathcal{E} = \{e_i\}_{i=1,\dots,N_\mathcal{E}}$. Here, the $j$-th entity in $\mathcal{D}$ is represented as $s_j = \{e_j, (w_{j_1}, \dots, w_{j_k})\}$, where $e_j \in \mathcal{E}$ is the entity type and $(w_{j_1}, \dots, w_{j_k})$ is a word sequence, where the words are two-by-two different but not necessarily adjacent. It is important to note that sequence-labeling based methods assign a BIO-label to each word and predict adjacent words $(w_j, w_{j+1}, \dots)$ as entities, which is not suitable for real-world VrD-NER where the reading order issue exists.

### 3.2 Token Path Prediction for VrD-NER

In Token Path Prediction, we model VrD-NER as predicting paths within a graph of tokens. Specifically, for a given document $\mathcal{D}$, we construct a complete directed graph with the $N_\mathcal{D}$ tokens $\{w_i\}_{i=1,\dots,N_\mathcal{D}}$ in $\mathcal{D}$ as vertices. This graph consists of $n^2$ directed edges pointing from each token to each token. For each entity $s_j = \{e_j, (w_{j_1}, \dots, w_{j_k})\}$, the word sequence can be represented by a path $(w_{j_1} \to w_{j_2}, \dots, w_{j_{k-1}} \to w_{j_k})$ within the graph, referred to as a token path.

TPP predicts the token paths in a given graph by learning grid labels. For each entity type, the token paths of entities are marked by a $n * n$ grid label of binary values, where $n$ is the number of text tokens. In specific, for each edge in every token path, the pair of the beginning and ending tokens is labeled as 1 in the grid label, while others are labeled as 0. For example in Figure 3, the token pair *"(NAME, OF)"* and *"(OF, ACCOUNT)"* are marked 1 in the grid label. In this way, entity annotations can be represented as $N_\mathcal{E}$ grids of $n * n$ binary values.

The learning of grid labels is then treated as $N_\mathcal{E}$ binary classification tasks on $n^2$ samples, which can be implemented using any classification model. In TPP, we utilize document transformers to represent document inputs as token feature sequences, and employ Global Pointer (Su et al., 2022) as the classification model to predict the grid labels by binary classification. Following (Su et al., 2022), the weights are optimized by a class-imbalance loss to overcome the class-imbalance problem, as there are at most $n$ positive labels out of $n^2$ labels in each grid. During evaluation, we collect the $N_\mathcal{E}$ predicted grids, filtering the token pairs predicted to be positive. If there are multiple token pairs with same beginning token, we only

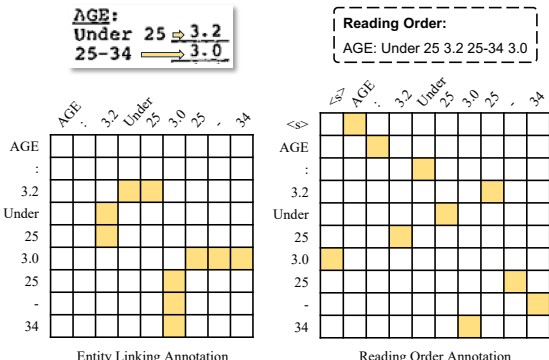

Figure 4: The grid label design of TPP for VrD-EL and VrD-ROP. For VrD-EL, *Under 25/25-34* is linking to *3.2/3.0* so each token between the two entities are linked. For VrD-ROP, the reading order is represented as a global path including all tokens starting from an auxiliary beginning token **, in which the linked token pairs are marked.

keep the pair with highest confidence score. After that, we perform a greedy search to predict token paths as entities. Figure 3 displays the overall procedure of TPP for VrD-NER.

In general, TPP is a simple and easy-to-implement solution to address the reading order issue in real-world VrD-NER.

### 3.3 Token Path Prediction for Other Tasks

We have explored the capability of TPP to address various VrD tasks, including VrD-EL and VrD-ROP. As depicted in Figure 4, these tasks are addressed by devising task-specific grid labels for TPP training. For the prediction of VrD-EL, we gather all token pairs between every two entities and calculate the mean logit score. Two entities are predicted to be linked if the mean score is greater than 0. In VrD-ROP, we perform a beam search on the logit scores, starting from the auxiliary beginning token to predict a global path linking all tokens. The feasibility of TPP on these tasks highlights its potential as a universal solution for VrD tasks.

## 4 Revised Datasets for Scanned VrD-NER

In this section, we introduce FUNSD-r and CORD-r, the revised VrD-NER datasets to reflect the real-world scenarios of NER on scanned VrDs. We first point out the existing problems of popular benchmarks for NER on scanned VrDs, which indicates the necessity of us to build new benchmarks. We then describe the construction of new datasets,

including selecting adequate data resources and the annotating process.

### 4.1 Motivation

FUNSD (Jaume et al., 2019) and CORD (Park et al., 2019) are the most popular benchmarks of NER on scanned VrDs. However, these benchmarks are biased towards real-world scenarios. In FUNSD and CORD, segment layout annotations are aligned with labeled entities. The scope of each entity on the document layout is marked by a bounding box that forms the segment annotation together with each entity word. We argue that there are two problems in their annotations that render them unsuitable for evaluating current methods. First, these benchmarks do not reflect the reading order issue of NER on scanned VrDs, as each entity corresponds to a continuous span in the model input. In these benchmarks, each segment models a continuous semantic unit where words are correctly ordered, and each entity corresponds exactly to one segment. Consequently, each entity corresponds to a continuous and front-to-back token span in the model input. However, in real-world scenarios, entity mentions may span across segments, and segments may be disordered, necessitating the consideration of reading order issues. For instance, as shown in Figure 5, the entity *"Sample Requisition [Form 02:02:06]"* is located in a chart cell spanning multiple rows; while the entity is recognized as two segments by the OCR system since OCR-annotated segments are confined to lie in a single row. Second, the segment layout annotations in current benchmarks vary in granularity, which is inconsistent with real-world situations. The scope of segment in these benchmarks ranges from a single word to a multi-row paragraph, whereas OCR-annotated segments always correspond to words within a single row.

Therefore, we argue that a new benchmark should be developed with segment layout annotations aligned with real-world situations and entity mentions labeled on words.

### 4.2 Dataset Construction

As illustrated above, current benchmarks cannot reflect real-world scenarios and adequate benchmarks are desired. That motivates us to develop new NER benchmarks of scanned VrDs with real-world layout annotations.

We achieve this by reannotating existing benchmarks and select FUNSD (Jaume et al., 2019)

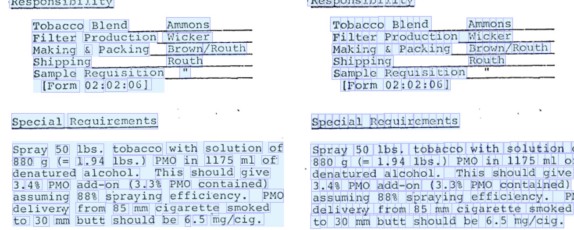

Figure 5: A comparison of the original (left) and revised (right) layout annotation. The revised layout annotation reflects real-world situations, where (1) character-level position boxes are annotated rather than word-level boxes, (2) every segment lies in a single row, (3) spatially-adjacent words are recognized into one segment without considering their semantic relation, and (4) missing annotation exists.

and CORD (Park et al., 2019) for the following reasons. First, FUNSD and CORD are the most commonly used benchmarks on VrD-NER as they contains high-quality document images that closely resemble real-world scenarios and clearly reflect the reading order issue. Second, the annotations on FUNSD and CORD are heavily biased due to the spurious correlation between its layout and entity annotations. In these benchmarks, each entity exactly corresponds to a segment, which leaks the ground truth entity labels during training by the 2D positional encoding of document transformers, as tokens of the same entity share the same segment 2D position information. Due to the above issues, we reannotate the layouts and entity mentions on the document images of the two datasets. We automatically reannotate the layouts using an OCR system, and manually reannotate the named entities as word sequences based on the new layout annotations to build the new datasets. The proposed FUNSD-r and CORD-r datasets consists of 199 and 999 document samples including the image, layout annotation of segments and words, and labeled entities. The detailed annotation pipeline and statistics of these datasets is introduced in Appendix A. We publicly release the two revised datasets at GitHub.[1]

## 5 Experiments

### 5.1 Datasets and Evaluation

The experiments in this work involve VrD-NER, VrD-EL, and VrD-ROP. For VrD-NER, the experiments are conducted on the FUNSD-r and CORD-r

[1] https://github.com/chongzhangFDU/TPP

datasets proposed in our work. The performance of methods are measured by entity-level F1 on these datasets. Noticing that previous works on FUNSD and CORD have used word-level F1 as the primary evaluation metric, we argue that this may not be entirely appropriate, as word-level F1 only evaluates the accuracy of individual words, yet entity-level F1 assesses the entire entity, including its boundary and type. Compared with word-level F1, entity-level F1 is more suitable for evaluation on NER systems in pratical applications. For VrD-EL, the experiments are conducted on FUNSD and the methods are measured by F1. For VrD-ROP, the experiments are conducted on ReadingBank. The methods are measured by Average Page-level BLEU (BLEU for short) and Average Relative Distance (ARD) (Wang et al., 2021b).

### 5.2 Baselines and Implementation Details

For VrD-NER, we compare the proposed TPP with sequence-labeling methods that adopts document transformers integrated with token classification heads. As these methods suffer from the reading order issue, we adopt a VrD-ROP model as the pre-processing mechanism to reorder the inputs, mitigating the impact of the reading order issue. We adopt LayoutLMv3-base (Huang et al., 2022) and LayoutMask (Tu et al., 2023) as the backbones to integrate with TPP or token classification heads, as they are the SOTA document transformers in base-level that use "vision+text+layout" and "text+layout" modalities, respectively. For VrD-EL and VrD-ROP, we introduce the baseline methods with whom we compare the proposed TPP in Appendix B. The implementation details of experiments are illustrated in Appendix C.

### 5.3 Evaluation on VrD-NER

In this section, we display and discuss the performance of different methods on VrD-NER in real-world scenario. In all, the effectiveness of TPP is demonstrated both as an independent VrD-NER model, and as a pre-processing mechanism to reorder inputs for sequence-labeling models. By both means, TPP outperforms the baseline models on two benchmarks, across both of the integrated backbones. It is concluded in Table 1 that: (1) TPP is effective as an independent VrD-NER model, surpassing the performance of sequence-labeling methods on two benchmarks for real-world VrD-NER. Since sequence-labeling methods are adversely affected by the disordered inputs,

| Backbone | Method | Pre. | Cont. (%) | F1 |
|---|---|---|---|---|
| LayoutLMv3 | Sequence Labeling | None | 95.74 | 78.77 |
| | | LR | 95.53 | 78.37 |
| | | $TPP_R$ | **97.29** | 79.72 |
| | TPP | None | - | **80.40** |
| LayoutMask | Sequence Labeling | None | 95.74 | 77.10 |
| | | LR | 95.53 | 77.24 |
| | | $TPP_R$ | **97.29** | **80.70** |
| | TPP | None | - | 78.19 |

(a) VrD-NER on FUNSD-r

| Backbone | Method | Pre. | Cont. (%) | F1 |
|---|---|---|---|---|
| LayoutLMv3 | Sequence Labeling | None | 92.10 | 82.72 |
| | | $LR_C$ | 82.10 | 70.33 |
| | | $TPP_C$ | **92.43** | 83.24 |
| | TPP | None | - | **91.85** |
| LayoutMask | Sequence Labeling | None | 92.10 | 81.84 |
| | | $LR_C$ | 82.10 | 68.05 |
| | | $TPP_C$ | **92.43** | 81.90 |
| | TPP | None | - | **89.34** |

(b) VrD-NER on CORD-r

Table 1: The VrD-NER performance of different methods on FUNSD-r and CORD-r. Pre. denotes the pre-processing mechanism used to re-arrange the input tokens, where $LR_*$/$TPP_*$ denotes that input tokens are reordered by a LayoutReader/TPP-for-VrD-ROP model, LR and $TPP_R$ are trained on ReadingBank, and $LR_C$ and $TPP_C$ are trained on CORD. Cont. denotes the continuous entity rate, higher for better pre-processing mechanism. The best F1 score and the best continuous entity rates are marked in bold. Note that TPP-for-VrD-NER methods do not leverage any reading order information from ground truth annotations or pre-processing mechanism predictions.

their performance is constrained by the ordered degree of datasets. In contrast, TPP is not affected by disordered inputs, as tokens are arranged during its prediction. Experiment results show that TPP outperforms sequence-labeling methods on both benchmarks. Particularly, TPP brings about a +9.13 and +7.50 performance gain with LayoutLMv3 and LayoutMask on the CORD-r benchmark. It is noticeable that the real superiority of TPP for VrD-NER might be greater than that is reflected by the results, as TPP actually learns a more challenging task comparing to sequence-labeling. While sequence-labeling solely learns to predict entity boundaries, TPP additionally learns to predict the token order within each entity, and still achieves better performance. (2) TPP is a desirable pre-processing mechanism to reorder inputs for sequence-labeling models, while the other VrD-ROP models are not. According to Table 1, LayoutReader and TPP-for-VrD-ROP are evaluated as the pre-processing mechanism by the continuous entity rate of rearranged inputs. As a representative of current VrD-ROP models, LayoutReader does not perform well as a pre-processing mechanism, as the continuous entity rate of inputs decreases after the arrangement of LayoutReader on two disordered datasets. In contrast, TPP performs satisfactorily. For reordering FUNSD-r, $TPP_R$ brings about a +1.55 gain on continuous entity rate, thereby enhancing the performance of sequence-labeling methods. For reordering CORD-r, the desired reading order of the documents is to read row-by-row, which conflicts with the reading order of ReadingBank documents that are read column-

by-column. Therefore, for fair comparison on CORD-r, we train LayoutReader and TPP on the original CORD to be used as pre-processing mechanisms, denoted as $LR_C$ and $TPP_C$. As illustrated in Table 1, $TPP_C$ also improves the continuous entity rate and the predict performance of sequence-labeling methods, comparing with the LayoutReader alternative. Contrary to TPP, we find that using LayoutReader for pre-processing would result in even worse performance on the two benchmarks. We attribute this to the fact that LayoutReader primarily focuses on optimizing BLEU scores on the benchmark, where the model is only required to accurately order 4 consecutive tokens. However, according to Table 4 in Appendix A, the average entity length is 16 and 7 words in the datasets, and any disordered token within an entity leads to its discontinuity in sequence-labeling. Consequently, LayoutReader may occasionally mispredict long entities with correct input order, due to its seq2seq nature which makes it possible to generate duplicate tokens or missing tokens during decoding.

For better understanding the performance of TPP, we conduct ablation studies to determine the best usage of TPP-for-VrD-NER by configuring the TPP and backbone settings. The results and detailed discussions can be found in Appendix D.

## 5.4 Evaluation on Other Tasks

Table 2 and 3 display the performance of VrD-EL and VrD-ROP methods, which highlights the potential of TPP as a universal solution for VrD-IE tasks. For VrD-EL, TPP outperforms MSAU-PAF,

| Case | Ground Truth | Sequence-labeling | Token Path Prediction |
|------|--------------|-------------------|----------------------|
| Multi-row Entity | | | |
| Multi-column Entity | | | |
| Long Entity | | | |
| Entity Type Identification | | | |

Figure 6: Case study of Token Path Prediction for VrD-NER, where entities of different types are distinguished by color, and different entities of the same type are distinguished by the shade of color.

| Method | F1 |
|--------|-----|
| GNN+MLP (Carbonell et al., 2021) | 39 |
| SPADE (Hwang et al., 2021) | 41.3 |
| Doc2Graph (Gemelli et al., 2022) | 53.36 |
| LayoutXLM (Xu et al., 2021b) | 54.83 |
| SERA (Zhang et al., 2021) | 65.96 |
| BROS (Hong et al., 2022) | 71.46 |
| MSAU-PAF (Dang et al., 2021) | 75 |
| TPP | **79.23** |

Table 2: The VrD-EL performance of different methods on FUNSD. The best result is marked in bold.

a competitive method, by a large margin of 4.23 on F1 scores. The result suggests that the current labeling scheme for VrD-EL provides adequate information for models to learn from. For VrD-ROP, TPP shows several advantages comparing with the SOTA method LayoutReader: (1) TPP is robust to input shuffling. The performance of TPP remains stable when train/evaluation inputs are shuffled as TPP is unaware of the token input order. However, LayoutReader is sensitive to input order and its performance decreases sharply when evaluation inputs are shuffled. (2) In principle, TPP is more suitable to act as a pre-processing reordering mechanism compared with LayoutReader. TPP surpasses LayoutReader by a significant margin on ARD among six distinct settings, which is mainly attributed to the paradigm differences. Specifically, TPP-for-VrD-ROP guarantees to predict a permutation of all tokens as the reading order, while LayoutReader predicts

| Order | Method | Avg. Page-level BLEU (%) | | |
|-------|--------|---------|---------|---------|
| | | $r$=100% | $r$=50% | $r$=0% |
| OCR | LayoutReader | 97.65 | 97.88 | **98.19** |
| | TPP | **98.18** | **98.13** | 98.16 |
| Shfl. | LayoutReader | 97.72 | 97.70 | 17.83 |
| | TPP | **98.16** | **98.09** | **98.12** |

(a) The average page-level BLEU of methods (higher is better).

| Order | Method | ARD | | |
|-------|--------|---------|---------|---------|
| | | $r$=100% | $r$=50% | $r$=0% |
| OCR | LayoutReader | 2.50 | 2.24 | 1.75 |
| | TPP | **0.29** | **0.35** | **0.37** |
| Shfl. | LayoutReader | 2.48 | 2.46 | 72.94 |
| | TPP | **0.37** | **0.39** | **0.39** |

(b) The average relative distance of methods (lower is better).

Table 3: The VrD-ROP performance of different methods on ReadingBank. For the Order setting, OCR denotes that inputs are arranged left-to-right and top-to-bottom in evaluation. Shfl. denotes that inputs are shuffled in evaluation. $r$ is the proportion of shuffled samples in training. The best results are marked in bold.

the reading order by generating a token sequence and carries the risk of missing some tokens in the document, which results in a negative impact the ARD metric, and also make LayoutReader unsuitable for use as a pre-processing reordering mechanism. (3) TPP surpasses LayoutReader among five out of six settings on BLEU. TPP is unaware of the token input order, and the performance of it among different settings is influenced only by random aspects. Although

LayoutReader has a higher performance in the setting where train and evaluation samples are both not shuffled, it is attributed to the possible overfitting of the encoder of LayoutReader on global 1D signals under this setting, where global 1D signals strongly hint the reading order.

## 5.5 Case Study

For better understanding the strength and weakness of TPP, we conduct a case study for analyzing the prediction results by TPP in VrD-NER. As discussed, TPP-for-VrD-NER make predictions by modeling token paths. Intuitively, TPP should be good at recognizing entity boundaries in VrDs. To verify this, we visualize the predict result of interested cases by TPP and other methods in Figure 6. According to the visualized results, TPP is good at identifying the entity boundary and makes accurate prediction of multi-row, multi-column and long entities. For instance, in the Multi-row Entity case, TPP accurately identifies the entity *"5/3/79 10/6/80, Update"* in complex layouts. In the Multi-column Entity case, TPP identifies the entity *"TOTAL 120,000"* with the interference of the injected texts within the same row. In the Long Entity case, the long entity as a paragraph is accurately and completely extracted, while the sequence-labeling method predicts the paragraph as two entities.

Nevertheless, TPP may occasionally misclassify entities as other types, resulting in suboptimal performance. For example, in the Entity Type Identification case, TPP recognizes the entity boundaries of *"SEPT 21"* and *"NOV 9"* but fails to predict their correct entity types. This error can be attributed to the over-reliance of TPP on layout features while neglecting the text signals.

## 6 Conclusion and Future Work

In this paper, we point out the reading order issue in VrD-NER, which can lead to suboptimal performance of current methods in practical applications. To address this issue, we propose Token Path Prediction, a simple and easy-to-implement method which models the task as predicting token paths from a complete directed graph of tokens. TPP can be applied to various VrD-IE tasks with a unified architecture. We conduct extensive experiments to verify the effectiveness of TPP on real-world VrD-NER, where it serves both as an independent VrD-NER model and as a pre-processing mechanism to reorder model inputs. Also, TPP achieves SOTA performance on VrD-EL and VrD-ROP tasks. We also propose two revised VrD-NER benchmarks to reflect the real situations of NER on scanned VrDs. In future, we plan to further improve our method and verify its effectiveness on more VrD-IE tasks. We hope that our work will inspire future research to identify and tackle the challenges posed by VrD-IE in practical scenarios.

## Limitations

Our method has the following limitations:

1. Relatively-high Computation Cost: Token Path Prediction involves the prediction of grid labels, therefore is more computational-heavily than vanilla token classification, though it is still a simple and generalizable method for VrD-IE tasks. Towards training on FUNSD-r, LayoutLMv3/LayoutMask with Token Path Prediction requires 2.45x/2.75x longer training time and 1.56x/1.47x higher peak memory occupancy compared to vanilla token classification. The impact is significant when training Token Path Prediction on general-purpose large-scale datasets.
2. Lack of Evaluation on Real-world Benchmarks: To illustrate the effectiveness of Token Path Prediction on information extraction of real-world scenarios, we rely on adequate downstream datasets. However, benchmarks is still missing of other IE tasks than NER, and of other languages than English. We appeal for more works to propose VrD-IE and VrDU benchmarks that reflect real-world scenarios, and the demonstration could be more concrete of our method to be capable of various VrD tasks, be compatible with document transformers of different modalities and multiple languages, and be open to future work of layout-aware representation models.

## Acknowledgements

The authors wish to thank the anonymous reviewers for their helpful comments. This work was partially funded by National Natural Science Foundation of China (No.62206057,61976056,62076069), Shanghai Rising-Star Program (23QA1400200), Natural Science Foundation of Shanghai (23ZR1403500), Program of Shanghai Academic Research Leader under grant 22XD1401100.

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

## A  The Annotation Pipeline and Statistics of Revised Datasets

The annotation pipeline is introduced as follows. First, we collect the document images of FUNSD and CORD, and re-annotate the layout automatically with an OCR system. We choose PP-OCRv3 (Li et al., 2022a) OCR engine as it is the SOTA solution of general-purpose OCR and can extend to multilingual settings. We operate each document image of FUNSD and CORD, keeping layout annotations with more than 0.8 confidence, and filtering document images with more than 20 valid words. After that, we manually annotate entity mentions on the new layouts as word sequences, based on the original entity annotations. Entity mentions that cannot match a word sequence on the new layout are deprecated. After the annotation, we divide the train/validation/test splits according to the original settings to obtain the revised datasets.

In all, the proposed FUNSD-r and CORD-r datasets consists of 199 and 999 document samples including the image, layout annotation of segments and words, and labeled entities. Table 4 lists the detailed summary statistics of the proposed datasets. The total number of segments, words, entities and entity types, the average length of segments and entities, and the sample number of data splits are displayed. Additionally, the continuous entity rate is introduced as the rate of entity whose tokens are continuous in the order of segment words. When fed into model, the continuous entities correspond to continuous token spans within inputs and are able to be predicted by sequence-labeling methods. Therefore, the continuous entity rate indicates the ordered degree of the document layouts in the dataset.

## B  Baselines for VrD-EL and VrD-ROP

For VrD-EL, we compare the proposed TPP with the following strong baseline models:
- GNN-based document encoders: GNN+MLP (Carbonell et al., 2021) is the pioneer to adopt GNN as the document encoder for better modeling of document inputs. Doc2Graph (Gemelli et al., 2022) is an enhanced GNN-based document encoder for VrD tasks.
- Transformer-based document encoders: LayoutXLM (Xu et al., 2021b) is a pre-trained document encoder for multilingual document inputs. BROS (Hong et al., 2022) is a pre-trained document encoder focuses on achieving better

| Case | Disordered Layout | Desired Reading Order | Reading Order by OCR Result |
|---|---|---|---|
| Multi-row Content | | 本年阶梯累计用水量 21 | 本年阶梯累计 21 用水量 |
| Seal with Askew or Overlapped Texts | | 当日当次 执收单位（盖章）
湖南省高速公路集团有限公司
票证专用章 | 湖南省高速公路集团 当日当次
有限公司票证专用章
执收单位（盖章） |
| Print Distortion | | 车道：X08 收费员：
车型：01 金额：13 元
入口站：株洲北 出口站：湘潭北 | X08 收费员：车道：13 01 元
车型：金额：株洲北 湘潭北
入口站：出口站： |
| Design Layout with Various Font Sizes | | 商品标价签
品名：利群（红） | 商品标价签
利群（红）品名： |
| Skewness due to Photography | | 年 月
品名 沫煤 规格 露天矿
毛重 68040 皮重 16980 | 年 月
沫煤 品名 规格 露天矿
皮重 毛重 68040 16980 |
| Semantic-driven Reading Order | | 承运单位 货达 派车证号 46525
品名 中硫焦精煤
单位 吨 毛重 43.22 | 派车 承运 证号 单位 货达 46525
毛重 单位 名 品
吨 中硫焦精煤 43.22 |

Figure 7: Examples of disordered layouts in real-world scenarios. These examples are taken from (Yu et al., 2023).

| Dataset | # of Segments | # of Words | Avg. Length of Segment | # of Entities | Avg. Length of Entity | Cont. (%) | # of Entity Types | # of Samples (Train/Val/Test) |
|---|---|---|---|---|---|---|---|---|
| **FUNSD-r** | 10,091 | 166,040 | 16.45 | 7,924 | 15.21 | 95.74 | 3 | 149/-/50 |
| **CORD-r** | 12,582 | 123,153 | 7.55 | 12,582 | 9.20 | 92.10 | 30 | 799/100/100 |

Table 4: Statistics of the proposed datasets. Cont. denotes the continuous entity rate, as the rate of entity whose tokens are continuous in the order of segment words. Higher continuous entity rate indicates to more orderly layouts.

performance on key information extraction tasks.

- Parsing-based methods: SPADE (Hwang et al., 2021) treats VrD-EL as spatial-aware dependency parsing problem and adopts a spatial-enhanced language model to address the task. SERA (Zhang et al., 2021) also treats the task as dependency parsing, and adopts a biaffine parser together with a document encoder to address the task.
- Detection-based methods: MSAU-PAF (Dang et al., 2021) tackles the task by integrating the MSAU architecture for detection (Dang and Nguyen, 2021) and the PIF-PAF mechanism for link prediction (Kreiss et al., 2019).

For VrD-ROP, we compare the proposed TPP with LayoutReader (Wang et al., 2021b), which is a sequence-to-sequence model achieving strong performance on VrD-ROP. We refer to the reported performance in the original papers of these baseline methods in Table 2 and 3. The backbone of all compared methods are of base-level.

## C  Implementation Details

The experiments in this work are divided into two parts, as experiments on VrD-NER and on other tasks.

For VrD-NER, we adopt LayoutLMv3-base (Huang et al., 2022) and LayoutMask (Tu et al., 2023) as the backbones to integrate with Token Path Prediction. The maximum sequence length of textual tokens for both of them is 512. We use Global 1D and Segment 2D position information in LayoutLMv3, and Local 1D and Segment 2D in LayoutMask, according to the preferred settings. We adopt positional residual linking and multi-dropout to improve the efficiency and robustness in training TPP. Positional residual linking adds the 1D positional embeddings of each token to the

backbone outputs to enhance 1D position signals that is crucial to the task. Multi-dropout duplicates the last fully-connect layer of TPP. Following the setting of (Inoue, 2019), the duplicated layers are trained with different dropout noises and their weights are shared, which enhances model robustness towards randomness. The effectiveness of these mechanisms are discussed in Appendix D. In fine-tuning, we generally follow the original setting of previous VrD-NER works (Huang et al., 2022; Tu et al., 2023). We use an Adam optimizer, with 1% linear warming-up steps, a 0.1 dropout rate, and a 1e-5 weight decay. For Token Path Prediction, the learning rate is searched from {3e-5, 5e-5, 8e-5}. On FUNSD, the best learning rate is 5e-5/5e-5 for LayoutLMv3/LayoutMask, while on CORD the learning rate is 3e-5/8e-5, respectively. In fine-tuning the comparing token classification models, we all set the learning rate to 5e-5. We adopt positional residual linking and multi-dropout in Token Path Prediction. We fine-tune FUNSD-r and CORD-r by 1,000 and 2,500 steps on 8 Tesla A100 GPUs, respectively, with a batch size of 16. The maximum number of decoded entities is limited to 100.

Besides to VrD-NER, we conduct experiments on VrD-EL and VrD-ROP adopting LayoutMask backbone. For VrD-EL, TPP-for-VrD-EL is fine-tuned by 1000 steps with a learning rate of 8e-5 and a batch size of 16, with the learning rate of the global pointer weights is set as 10x for better convergence. For VrD-ROP, TPP-for-VrD-ROP is fine-tuned by 100 epochs with a learning rate of 5e-5 and a batch size of 16, following the settings of (Wang et al., 2021b). The beam size is set to 8 during decoding. For all experiments, we choose the model checkpoint with the best performance on the validation set, and report its performance on the test set.

## D  Ablation Studies to TPP on VrD-NER

For better understanding of TPP on VrD-NER, we propose two questions: (1) from which setting TPP benefits most of positional encoding of backbones, and (2) by what means positional residual linking and multi-dropout be helpful to TPP. We conduct the ablation studies to answer the questions.

For the first question, we alter the choices of 1D and 2D positional encoding of backbones, and the results are reported in Table 5a. For FUNSD-r, the preferred positional encoding setting of backbones

| Backbone | Position | | FUNSD-r | CORD-r |
|---|---|---|---|---|
| | 1D | 2D | | |
| LayoutLMv3 | Global | Segment | 80.40 | 91.85 |
| | None | Segment | 74.15 | 89.91 |
| | Global | Word | 77.86 | 90.45 |
| LayoutMask | Local | Segment | 78.19 | 89.34 |
| | Global | Segment | 66.24 | 87.36 |
| | None | Segment | 67.81 | 82.27 |
| | Local | Word | 75.96 | 89.45 |

(a) Ablation study on different 1D and 2D positional encoding choices of backbones.

| Backbone | Method | FUNSD-r | CORD-r |
|---|---|---|---|
| LayoutLMv3 | TPP | 80.40 | 91.85 |
| | (w/o mul.) | 79.61 | 90.99 |
| | (w/o mul. & res.) | 78.93 | 91.14 |
| LayoutMask | TPP | 78.19 | 89.34 |
| | (w/o mul.) | 75.34 | 87.96 |
| | (w/o mul. & res.) | 65.53 | 89.55 |

(b) Ablation study on the design of Token Path Prediction. Positional residual linking and multi-dropout are abbreviated as res. and mul., respectively.

Table 5: Ablation studies to TPP-for-VrD-NER on position encoding choices of backbones and model design.

brings about the best performance. Although TPP is unaware of the order of tokens, the incorporation of better 1D positional signals can enhance the contextualized representation generated by the backbones, leading to improved performance. For CORD-r, we note that in some occasions, using word-level 2D position information brings about better performance. This is because short entities are the majority in CORD-r, especially entities with only one char, such as numbers and symbols. For these entities, word-level 2D position information brings about a direct hint to the model in training and prediction, resulting in better performance.

For the second question, we conduct an ablation study by removing the positional residual linking and multi-dropout of TPP. We observe from Table 5b that: (1) Positional residual linking and multi-dropout are both essential due to their effectiveness on the results of FUNSD-r. (2) Positional residual linking enhances the 1D position signals in model inputs. Comparing with vanilla TPP, TPP with positional residual linking behaves better on FUNSD-r but slightly worse on CORD-r. This is because segments in CORD-r are relatively short, when inputs are disordered, 1D signals are much more noisy and negatively affects model performance

since they do not provide sufficient information. The results verify the function of positional residual linking as a enhancement mechanism of 1D signals. (3) Multi-dropout enhances model robustness to random aspects. TPP outperforms its variant without multi-dropout among different backbones and different benchmarks.