# OpenReview forum: "Reading Order Matters: Information Extraction from Visually-rich Documents by Token Path Prediction"
_EMNLP/2023/Conference — EMNLP 2023 Main_

### Official Review · Reviewer_crwh · 2023-07-31

**Typos Grammar Style And Presentation Improvements:** I have not found any, only in line 10…
**Soundness:** 5

**Excitement:**

5: Transformative: This paper is likely to change its subfield or computational linguistics broadly. It should be considered for a best paper award. This paper changes the current understanding of some phenomenon, shows a widely held practice to be erroneous in someway, enables a promising direction of research for a (broad or narrow) topic, or creates an exciting new technique.

**Missing References:**

This work is somehow related: [1] Garncarek, Ł. et al. (2021). LAMBERT: Layout-Aware Language Modeling for Information Extraction. In: Lladós, J., Lopresti, D., Uchida, S. (eds) Document Analysis and Recognition – ICDAR 2021. ICDAR 2021. Lecture Notes in Computer Science(), vol 12821. Springer, Cham. https://doi.org/10.1007/978-3-030-86549-8_34

I hope the authors can compare and it can add additional interesting perspective to the paper.

**Paper Topic And Main Contributions:**

At first, the Authors pointed out problems with current datasets and 2 benchmarks on visually rich document recognition tasks (NER, EL< ROP).
Then, they propose a new method for token path prediction and explain their approach and revised datasets for those tasks.


**Questions For The Authors:**

1) How could you add visual channel/input to your model? Would it benefit?
2) Can we overcome the constructing graphs of tokens on the plain page by utilising visual embeddings?
3) How is your work related to the LAMBERT model and its input types, described in [1]?
4) Can you explain Figure 5? I cannot see much difference; maybe you can point out particular bounding boxes or describe them in the figure description.
5) Please, add the continuation of the sentence in line 1021 in the appendix.

[1] Garncarek, Ł. et al. (2021). LAMBERT: Layout-Aware Language Modeling for Information Extraction. In: Lladós, J., Lopresti, D., Uchida, S. (eds) Document Analysis and Recognition – ICDAR 2021. ICDAR 2021. Lecture Notes in Computer Science(), vol 12821. Springer, Cham. https://doi.org/10.1007/978-3-030-86549-8_34


**Reasons To Accept:**

- the Authors carefully analysed the topic and challenges in visually rich document recognition tasks.
- then, they carefully revised the existing two datasets (I.e. FUNSD, CORD) and re-annotated them.
- then, they proposed a new solution that defeated other benchmarks.
- the paper is very clear, with vivid examples and descriptions even for less-informed readers - I appreciate it very much!

**Reasons To Reject:**

I have not found any.

**Reproducibility:**

5: Could easily reproduce the results.

**Reviewer Confidence:**

4: Quite sure. I tried to check the important points carefully. It's unlikely, though conceivable, that I missed something that should affect my ratings.

---

> ### Author Rebuttal · Authors · 2023-08-28
>
> We thank you for your reviews and your appreciation to our work.
>
> We also thank you for recommending a related work, LAMBERT [1], which is an important work of self-supervised pre-trained document transformer with text+layout inputs at the era it is proposed. It achieves great performance on several VrD-IE tasks.  We thank you for sharing this interesting work, and we will add the citation of this work in the revised version.
>
> Your concerns are addressed as follows:
>
>
>
> **Q1:** About Question 1, "would it benefit to add visual inputs"
>
> **A1:** Token Path Prediction (TPP) is a VrD-NER task head and is compatible with commonly used document transformers. Whether to use visual inputs is decided by the upstream document transformers but not TPP. For your question, leveraging visual inputs would be beneficial as in Table 1, those results of LayoutLMv3 are generally better than those of LayoutMask.
>
>
>
> **Q2:** About Question 2, "overcoming the construction of token graphs by adding visual inputs"
>
> **A2:** As illustrated above, we can choose whether to use visual inputs, but merely using visual inputs is not enough to solve VrD-NER with reading order issue. The solution relies on the modeling and decoding strategy of TPP.
>
>
>
> **Q3:** About Question 3, "the relation between our work and LAMBERT and its input types"
>
> **A3:** As stated in the response of Question 1, TPP can be integrated together with LAMBERT, where LAMBERT serves as the upstream representation model of document inputs, and TPP acts as the downstream task model. Regrettably, due to the limited time of author response period, we are unable to present experiment results to demonstrate the effectiveness of LAMBERT+TPP on the VrD tasks. However, considering that LAMBERT shares the same input modalities and possesses a similar size of LayoutMask, the performance gain of LAMBERT by incorporating TPP should be comparable to the results on LayoutMask.
>
>
>
> **Q4:** About Question 4, "precise explanation of Figure 5"
>
> **A4:** We apologize for the lack of clarity, and we will reorganize the content of Section 4.2 to illustrate this figure more clearly in the revised version. There are basically 3 differences between the original and revised layouts:
>
> 1. The original layout annotates word-level position boxes, yet the revised layout annotates character-level position boxes. The latter is more in line with the outputs generated by OCR systems.
> 2. In the original layout, segments may span multiple rows, but in the revised layout, every segment lies in a single row. For example, the long paragraph "Spray 50 lbs ... 6.5 mg/cig. " is annotated as a whole segment in the original layout, but split to 7 segments in the revised layout, with each row being annotated as an individual segment.
> 3. The segment annotations on the original layout are manually labeled, ensuring a generally precise alignment with the semantic units of texts. However, in the revised layout, the segment annotations are labeled by text-region detection models and do not consider the semantic meaning of texts. As a result, (1) some unrelated texts may be grouped together in the same segment, such as "Filter Production" and "Wicker"; and (2) some short texts, like the symbol " on the right, may not be recognized by the OCR system.
>
>
>
> **Q5:** About Question 5, "one sentence is unfinished in the Appendix"
>
> **A5:** We thank you for pointing out this mistake and sincerely apologize for it. This mistake would be corrected in the revised version. The complete sentence at line 1021 should be, "This is because segments in CORD-r are relatively short, when inputs are disordered, 1D signals are much noisier and could not provide sufficient information."
>
>
>
> Reference:
>
> 1. Garncarek, Łukasz, et al. "Lambert: Layout-aware language modeling for information extraction." International Conference on Document Analysis and Recognition. Cham: Springer International Publishing, 2021.

---

### Official Review · Reviewer_bY5f · 2023-08-05

**Soundness:** 3

**Excitement:**

3: Ambivalent: It has merits (e.g., it reports state-of-the-art results, the idea is nice), but there are key weaknesses (e.g., it describes incremental work), and it can significantly benefit from another round of revision. However, I won't object to accepting it if my co-reviewers champion it.

**Paper Topic And Main Contributions:**

The paper proposes a method to deal with the reading order issue in the NER tasks on visually-rich documents. Also the authors present two revised benchmark datasets of NER tasks on scanned documents. Experimental results show the proposed method is efficient.

**Questions For The Authors:**

QA:  I think it would be interesting to see how the proposed the model preform on the original datasets (not the revised one) and comparing to the previous work.

**Reasons To Accept:**

The paper proposes a token path prediction method to dealing with the reading order issue in the visually-rich documents for NER tasks. The proposed method outperforms SOTA methods in various VrD tasks.

**Reasons To Reject:**

The experiments are conducted on the revised dataset, so there is no fair comparisons with the existing methods.

**Reproducibility:**

3: Could reproduce the results with some difficulty. The settings of parameters are underspecified or subjectively determined; the training/evaluation data are not widely available.

**Reviewer Confidence:**

3: Pretty sure, but there's a chance I missed something. Although I have a good feel for this area in general, I did not carefully check the paper's details, e.g., the math, experimental design, or novelty.

---

> ### Author Rebuttal · Authors · 2023-08-28
>
> We thank you for your reviews and your appreciation to our work. Your concerns are addressed as follows:
>
> **Q1:** About Reasons To Reject Point 1 and Question 1, "no fair comparisons with the existing methods"
>
> **A1:** It is unable to make fair comparisons between TPP and other methods on FUNSD and CORD due to the following reasons: (1) Other methods leverage the spurious correlation between segment layout and entity annotations, leading to falsely high performance. As discussed in Section 4.1, it is demonstrated that each entity corresponds exactly to one segment in FUNSD and CORD. The segment layout annotations of FUNSD and CORD leaks the entity boundary information, and previous models take advantage of this. (2) The task for TPP is harder to tackle VrD-NER. TPP is unaware of the input order of tokens. Therefore, in addition to detecting entity boundaries like other methods, TPP needs to arrange the token order within the entity as well.
>
> However, **fair comparisons on public benchmarks are conducted** for VrD-EL and VrD-ROP tasks, where TPP outperforms other strong baseline methods and achieves SOTA performance, even without any specialized design or optimization for these particular tasks. We believe that this is enough to prove the competitiveness of TPP among other methods.
>
>
>
>
>
> Additionally, there are two clarifications that need to be addressed:
>
> 1. Re-emphasizing our motivation
>
> - In response to your review comments, we would like to re-emphasize the main motivation behind our paper, to aware of the difficulties of VrD tasks in real-world. Throughout the article, we have consistently emphasized the criticality of the reading order issue in real-world VrD tasks, particularly VrD-NER. To our best knowledge, this issue has not been adequately addressed to date. Besides, we note that current public benchmarks (FUNSD, CORD, ...) fail to highlight this problem, as they only annotate entity labels on continuous word sequences. Consequently, existing models are affected that they focus on improving performance on these benchmarks, but none have successfully resolved the reading order issue.
>
> - Our work aims to draw the attention of researchers to this phenomenon, and inspire the development of new VrD-NER models to meet the need of practical applications. To accomplish this, we re-formulate the task and introduce two benchmarks (FUNSD-r and CORD-r) as well as one baseline method (TPP). By doing so, we hope to foster discussions within the field and encourage the development of solutions for real-world VrD tasks, but not overfitting benchmarks.
>
> 2. About the concern of the reproducibility of our work
>
> - We notice that this review addresses some concern about the reproducibility of our work. The two proposed datasets, codes and model checkpoints would be released after this paper is accepted.
>
> - Besides, all experimental details have been thoroughly presented in Appendix C to demonstrate that our experimental results are trustworthy and reproducible, which we believe reflects our utmost dedication and precision.

---

### Official Review · Reviewer_kJnu · 2023-08-12

**Soundness:** 4

**Excitement:**

4: Strong: This paper deepens the understanding of some phenomenon or lowers the barriers to an existing research direction.

**Paper Topic And Main Contributions:**

The authors point out and provide multiple examples of the shortcomings of the traditional sequence tagging scheme when applied to entity extraction from visually rich document, such as reliance on correct reading order, which is hard to obtain of realistic OCR.

They propose an alternative to sequence labeling for NER, token path prediction (TPP) that is invariant to reading order where all tokens in the document are treated as nodes of a graph and entities correspond to paths through this graph.

To do TPP, they modify the token standard classification head for binary grid label prediction.

They employ their approach as a standalone system for NER and entity linking (EL) as well as, reading order prediction (ROP) for preprocessing the input for standard sequence labeling.

Finally, they showcase performance gains on modified versions of FUNSD and CORD to include raw reading order errors and entity discontinuities. They show moderate to dramatic improvement over standard LayoutLMv3 and a collection of base lines including ones specifically designed to handle reading order issues.

**Questions For The Authors:**

Question A: Any insight as to why using LayoutReader for preprocessing results in even lower continuity and F1 scores, than when using standard sequence labeling LayoutLM on unmodified dataset.

Question B: In table 3 it appears different shuffling ratios r apply to OCR-ordered input, which I understand is not shuffled. How can this be?

**Reasons To Accept:**

The proposal of a reading order invariant labeling scheme TPP for VDU is well motivated and is novel, as far as I can tell.

The authors' solution of using TPP as grid label prediction is simple but sound and offers a number of advantages including the added ability to perform entity linking. They make appropriate adjustments to the training, such as using class-imbalance loss to accommodate the new labeling scheme.

They make reasonable modifications to the annotations of the existing datasets to highlight the shortcomings of reading order dependence in the presence of OCR errors; and conduct a compelling evaluation of their approach in multiple settings (as a standalone NER and as preprocessing for standard sequence labeling), which shows significant gains of using their method across NER, EL, and ROP tasks.

Their ablation and case studies appear compelling as well.

**Reasons To Reject:**

While conceptually the TPP makes a lot of sense, its practical impact may be limited as entity continuity rates seem still relatively high even in modified datasets.

Regarding TPP training for reading order prediction, the original CORD dataset is used for training, which could be causing data leakage. (Margin of improvement for CORD-r seems suspiciously higher than for FUNSD)

**Reproducibility:**

4: Could mostly reproduce the results, but there may be some variation because of sample variance or minor variations in their interpretation of the protocol or method.

**Reviewer Confidence:**

4: Quite sure. I tried to check the important points carefully. It's unlikely, though conceivable, that I missed something that should affect my ratings.

**Typos Grammar Style And Presentation Improvements:**

Typos:

tokens. In specific, for each edge in every token 282

268 we modeling

---

> ### Author Rebuttal · Authors · 2023-08-28
>
> We thank you for your reviews and your appreciation to our work.
>
> We notice your suggestion that our work could also be named as token trajectory prediction (TTP), which we think is also reasonable. To maintain consistency within the ongoing OpenReview discussion, we retain the original name of our method as Token Path Prediction (TPP).
>
> Your concerns are addressed as follows:
>
>
>
> **Q1:** About Reasons To Reject Point 1, "the practical impact of TPP may be limited as entity continuity rates are relatively high in revised datasets"
>
> **A1:** In fact, TPP becomes even more useful in practical applications, as there would be massive cases of document layouts that exhibit intensive disorder, such as (1) information extraction on scanned cards/passports with multi-row contents, seals with askew texts, and registration forms and tables with print distortion, for document analysis; and (2) visual qa on infographics and hand-written documents for document understanding. Moreover, in these documents, the disordered entities are usually critical ones (eg. dates, seal texts, etc. ), and are difficult to be corrected by regular matching. **It is this practical reality that primarily drives our motivation.** In the revised version, we will provide more real-world document examples with severe disordered layouts to further support our claims.
>
> Besides, compared with real-world scanned documents, FUNSD-r and CORD-r have neat layouts and thus have higher entity continuity rates. Despite this advantage,  sequence-labeling methods on them still suffer from significant performance drop. This observation indicates the seriousness of the reading order issue in practical scenarios.
>
>
>
> **Q2:** About Reasons To Reject Point 2, "the possible data leakage in experiments"
>
> **A2:** By "Margin of improvement for CORD-r seems suspiciously higher than for FUNSD", we think you refer to the performance of "Document Transformer + None Pre-processing + TPP-NER", abbriviate as "None+TPP", which is 91.85/89.34 with two backbones, and is much higher than None+SL (82.72/81.84). However, **None+TPP does not leverage any ground truth reading order annotations of CORD**, indicating that its superior performance is not a result of data leakage. As illustrated in Section 5.5, we believe it is the ability of identifying the entity boundary of TPP that attributes to the superior performance of None+TPP.
>
> However, the comparison between LR+SL and TPP_C+SL is somehow unfair, as the latter is aware of the ground truth reading order annotations of CORD, while the former is not. Due to it, we finetuned LayoutReader on CORD, noted as "LR_C", adopt it as the pre-processing mechanism, and the results are as follows:
>
> - Entity continuity rate after LR_C pre-processing: 82.10.
> - The F1 score of LR_C+SL on CORD: 70.33 for LayoutLMv3, and 68.05 for LayoutMask.
>
> We will replace the original results of LR+SL in Table 1b by the above results in the revised version. Note that the newly-reported results are still lower than that without pre-processing mechanisms. This phenomenon is illustrated below.
>
>
>
> **Q3:** About Question A, "why using LayoutReader for preprocessing results in worse performance"
>
> **A3:** According to [1], LayoutReader primarily focuses on optimizing BLEU scores on the benchmark, where the model is only required to accurately order 4 consecutive tokens. However, according to Table 4 in the appendix of our paper, the average length of entities is 16 and 7 words in the datasets. In this case, any disordered token within an entity leads to discontinuity. Consequently, LayoutReader may occasionally mispredict long entities with correct input order, due to its seq2seq nature which makes it possible to generate duplicate tokens or missing tokens during decoding.
>
> To address this issue more clearly, we will reorganize the content and provide clearer illustrations of this phenomenon in the revised version.
>
>
>
> **Q4:** About Question B, "confusion on the settings of Table 3"
>
> **A4:** For Table 3, we adopt the identical settings as LayoutReader [1]: the rate r% implies that r% train samples are randomly ordered, while the rest train samples remain in OCR order; the abbreviation OCR and Shfl. denotes that the validation samples are in OCR/random order.
>
> Table 3 illustrates the superiority of TPP for reading order prediction, as its performance is generally better than LayoutReader, and is invariant of the shuffle rate r%.
>
>
>
> **Q5:** About suggestions on Typos, Grammar Style And Presentation Improvements
>
> **A5:** We thank you for pointing out the typos. These mistakes would be corrected in the revised version.
>
>
>
> References:
>
> 1. Wang, Zilong, et al. "Layoutreader: Pre-training of text and layout for reading order detection." arXiv preprint arXiv:2108.11591 (2021).

---

### Meta-Review · Area_Chair_DESd · 2023-09-15

**Recommendation:** 5

**Metareview:**

The paper pointed out problems with current datasets and 2 benchmarks on visually rich document recognition tasks (NER, EL< ROP). The authors propose a new method for token path prediction and explain their approach and revised datasets for those tasks. The proposal of a reading order invariant labeling scheme TPP for VDU is well motivated and is novel,

---

### Decision · Program_Chairs · 2023-10-07

**Decision:**

Accept-Main

**Comment:**

The paper pointed out problems with current datasets and 2 benchmarks on visually rich document recognition tasks (NER, EL< ROP). The authors propose a new method for token path prediction and explain their approach and revised datasets for those tasks. The proposal of a reading order invariant labeling scheme TPP for VDU is well motivated and is novel,